Endoplasmic reticulum stress in the pathogenesis of alcoholic liver disease

Na Man
Yang Xingbiao
Deng Yongkun
Yin Zhaoheng
Li Mingwei 2747311411@qq.com
Department of Pharmacy, The 926th Hospital of Joint Logistics Support Force of Chinese People’s Liberation Army , Kaiyuan, Yunan , China
Banerjee Priyanka
Electronic publication date: 2023 Nov 14
Publication date: 2023
Volume: 11
Electronic Location ID: e16398
Received 2023 Aug 2; Accepted 2023 Oct 12
Copyright: © 2023 Na et al.
Copyright year: 2023
Copyright holder: Na et al.
License: This is an open access article distributed under the terms of the Creative Commons Attribution License, which permits unrestricted use, distribution, reproduction and adaptation in any medium and for any purpose provided that it is properly attributed. For attribution, the original author(s), title, publication source (PeerJ) and either DOI or URL of the article must be cited.
License URL: https://creativecommons.org/licenses/by/4.0/

Keywords: Alcoholic liver disease, Endoplasmic reticulum stress, Unfolded protein response, Hepatocyte apoptosis

Funding: The authors received no funding for this work.

==============================
The endoplasmic reticulum (ER) plays a pivotal role in protein synthesis, folding, and modification. Under stress conditions such as oxidative stress and inflammation, the ER can become overwhelmed, leading to an accumulation of misfolded proteins and ensuing ER stress. This triggers the unfolded protein response (UPR) designed to restore ER homeostasis. Alcoholic liver disease (ALD), a spectrum disorder resulting from chronic alcohol consumption, encompasses conditions from fatty liver and alcoholic hepatitis to cirrhosis. Metabolites of alcohol can incite oxidative stress and inflammation in hepatic cells, instigating ER stress. Prolonged alcohol exposure further disrupts protein homeostasis, exacerbating ER stress which can lead to irreversible hepatocellular damage and ALD progression. Elucidating the contribution of ER stress to ALD pathogenesis may pave the way for innovative therapeutic interventions. This review delves into ER stress, its basic signaling pathways, and its role in the alcoholic liver injury.

Introduction

Alcoholic liver disease (ALD) is a significant concern for hepatologists, clinical researchers, and healthcare professionals involved in liver disease management, ranking among the most common liver afflictions worldwide. Its manifestations, resulting primarily from chronic excessive alcohol consumption, range from asymptomatic fatty degeneration and alcoholic steatohepatitis to more severe conditions such as alcoholic liver fibrosis, cirrhosis, and even alcohol-related hepatocellular carcinoma (Hyun et al., 2022). Notably, in 2019, approximately 25% of cirrhosis-related mortalities worldwide were linked to alcohol consumption (Huang et al., 2023).

The endoplasmic reticulum (ER) serves as the cellular hub for protein folding. Accumulation of improperly folded proteins within the ER instigates an uptick in misfolded protein levels. In response, the unfolded protein response (UPR) is initiated to restore ER equilibrium (Keerthiga, Pei & Fu, 2021). UPR can be activated through three pathways: IRE1α, ATF6α, and PERK pathways. Activation of the UPR pathways can increase the capacity of the ER and correct misfolded proteins. However, sustained ER stress can overwhelm the adaptive capacity of UPR, culminating in cellular apoptosis. It is noteworthy that alcohol-induced liver injuries are characterized by UPR activation, which is concomitant with pathologies like reactive oxygen species (ROS) generation and mitochondrial impairment (Xia et al., 2020).

Hepatocytes, the primary cellular units in the liver, fulfill intricate metabolic functions. These include plasma proteins synthesis and secretion, lipoproteins and very low-density lipoprotein discharges, cholesterol synthesis, and detoxification of foreign substances (Yoon & Kim, 2023). Owing to these multifarious roles, hepatocytes are replete with both smooth and rough ER. Figure 1 elucidates the association between ER stress and ALD. The functions of ER are well-established. However, the intricacies of how ALD-related perturbations in ER stress and homeostasis modulate ER functionality, and how such disruptions influence hepatocyte operations, remains enigmatic. ER stress manifestations are discernible across a spectrum of ALD cases (Xia et al., 2020). Existing studies allude to the potential of ER stress in inciting lipid metabolism anomalies, inflammation, and cell apoptosis (Yoon & Kim, 2023; Lebeaupin et al., 2018). Yet, the exact mechanisms whereby these anomalies induce hepatocyte injury and subsequently exacerbate ALD warrant comprehensive exploration.

Figure 1 Alcohol consumption influences ER stress and thus contributes to ALD.

This review, aimed at both budding researchers and seasoned experts in hepatology, cellular biology, and related fields, traces the origin of UPR under ER stress conditions, highlights the tendency of alcohol to prompt ER stress, and sheds light on ER stress’s central role in ALD progression.

Survey methodology

The PubMed database was used for related literature search using the keyword ‘alcoholic liver disease’, ‘endoplasmic reticulum stress’, ‘unfolded protein response’, ‘hepatocyte apoptosis’.

UPR pathways in hepatic ER stress

The liver stands as a pivotal organ, orchestrating crucial metabolic, secretory, and excretory functions. Hepatocytes in the liver are rich in ER to conduct processes such as the synthesis and secretion of very low-density lipoprotein and plasma proteins. In the liver, alcohol escalates oxidative stress, thereby impeding protein folding and modification and consequently triggering ER stress. ER stress further activates the UPR to maintain ER homeostasis. The activation of the UPR due to ER stress mainly depends on the activation of three pathways (Lebeaupin et al., 2018): (1) inositol-requiring enzyme 1 (IRE1), (2) activating transcription factor 6 (ATF6), and (3) protein kinase RNA-like ER kinase (PERK).

IRE1, a universally conserved UPR pathway present in both yeast and mammalian cells (Ren et al., 2021), is a dual-functional transmembrane protein possessing both serine/threonine kinase and ribonuclease activities. Its N-terminal domain, oriented towards the ER, is adept at detecting ER stress (Aghaei et al., 2020). Once ER stress is detected, IRE1 undergoes dimerization and autophosphorylation, thereby activating its RNase domain (Hughes & Mallucci, 2019). This aids in mitigating ER stress and enhancing protein folding and processing by activating the expression of downstream genes. Through the collaborative action of IRE1 and tRNA ligase RTCB, the transcription factor X-box binding protein (XBP1) undergoes cleavage and activated into the active spliced XBP1 (XBP1s) (Han & Kaufman, 2016). XBP1s promotes ER protein folding and secretion, enhances ER-associated degradation (ERAD), and fosters lipid synthesis (Hetz et al., 2011). Recent research has demonstrated that inhibitors that targeting the IRE1 signaling pathway effectively hinder the activation of fibroblasts induced by TGFβ in vitro. Subsequently, this inhibition mechanism has been found to lead to a marked reduction in liver fibrosis in vivo (Heindryckx et al., 2016).

PERK, a transmembrane protein with two structural domains: an N-terminal stress-sensing domain and a cytoplasmic kinase domain. During ER stress, PERK undergoes phosphorylation, leading to homodimer formation. To alleviate ER stress, PERK phosphorylates the alpha subunit of eukaryotic translation initiation factor 2 (eIF2α), inhibiting the assembly of the 80S ribosome and halting mRNA translation. Besides this inhibitory action, eIF2α also enhances the expression of the transcription factor ATF4. ATF4 regulates the expression of the growth arrest and DNA damage-inducible 34 (GADD34) protein. GADD34 functioning as a co-factor for phosphatases, responds to stress and promoting the dephosphorylation of eIF2α, thereby reinstating protein translation.

ATF6, another transmembrane protein, manifests in two isoforms two isoforms: ATF6α and ATF6β (Macke et al., 2023). Under ER stress, ATF6α migrates to the Golgi apparatus, undergoing cleavage to liberate the cytoplasmic fragment ATF6f. Subsequently, ATF6f relocates to the nucleus, regulating the cAMP response element and ER stress response elements. This regulation entails controlling downstream targets such as GRP78, ERAD-related proteins, and XBP1 (Walter et al., 2018). Studies have shown that ATF6 gene-knockout mice, upon being subjected to intraperitoneal injections with the ER stress inducer tunicamycin, exhibited a pronounced liver dysfunction and steatosis compared to their wild-type counterparts (Yamamoto et al., 2010).

Various pathways of UPR are illustrated in Fig. 2. Upon the onset of ER stress, cells activate the UPR via the pathways mentioned above. The UPR alleviates ER stress by reducing protein translation, increasing the expression of protein folding enzymes, and promoting protein degradation. However, chronic alcohol consumption can intensify ER stress. Persistent ER stress may exceed the adaptive threshold of UPR, leading to a further disturbance in hepatocyte physiology.

Figure 2 Diagram illustrating the three major unfolded protein response (UPR) pathways.

Alcohol consumption and ER stress

Ethanol consumption has been directly implicated in inducing ER stress. Excessive alcohol intake disturbs the functioning of the ER, resulting in hepatic manifestations such as steatosis and liver inflammation. The mechanisms through which alcohol consumption induces ER stress are described below.

Alcohol metabolism and reactive oxygen species: catalysts for ER stress

As shown in Fig. 1, ingested alcohol is oxidized to acetaldehyde by the alcohol dehydrogenase (ADH) enzyme, which is predominantly present in liver cells. This enzymatic reaction requires NAD+ as a coenzyme (Yoon & Kim, 2023). Acetaldehyde is subsequently converted to acetic acid by the mitochondrial aldehyde dehydrogenase (ALDH) enzyme. Notably, ALDH2 knockout mice exhibit a marked accumulation of acetaldehyde (Li et al., 2019). Acetaldehyde is known to promote protein oxidation, diminish SOD enzyme activity, and decrease the glutathione/oxidized GSH ratio (Farfán Labonne et al., 2009). Li et al. (2017) found that decreased expression levels of ADH and ALDH could suppress ROS generation in mice with alcohol-induced liver injury. Another pathway in ethanol metabolism is the microsomal ethanol-oxidizing system (MEOS), in which the cytochrome P450 enzyme CYP2E1 operates with the coenzyme NADPH (Cioarca-Nedelcu, Atanasiu & Stoian, 2021). Research indicates a significant increase in the ROS-generating enzymes CYP2E1 and NOX4 expression after exposing mice to ethanol acutely for 24 h (Chen et al., 2021). Although ethanol can be oxidized via the peroxisomal pathway dependent on hydrogen peroxide, it’s not the primary route for its metabolism (Singh, Osna & Kharbanda, 2017). When ethanol is metabolized via the ADH and MEOS pathways, a significant amount of NADH or NADP+ is generated, causing ROS accumulation and aggravating ER stress (Teschke, 2019). In addition, studies have revealed that alcohol consumption diminishes the activity of different antioxidant enzymes, such as superoxide dismutase, catalase, selenium (Se)-dependent glutathione peroxidase, glutathione reductase, and glutathione S-transferase. This decline is particularly pronounced in older mice, which exacerbates oxidative liver damage (Mallikarjuna et al., 2010; Shanmugam, Mallikarjuna & Reddy, 2011). Moreover, reduced levels of vitamins E and C have been observed in patients with ALD (Masalkar & Abhang, 2005), highlighting the diminished antioxidative capacities in such individuals.

The cytoplasm creates a reducing setting, with a proportion of reduced glutathione to oxidized glutathione that exceeds 50:1, this ratio within the ER lumen is only 3:1 (Yap & Say, 2011). This oxidizing ambiance in the ER lumen supports the formation of protein disulfide bonds and facilitates oxidative protein folding (Zeeshan et al., 2016). Enzymes such as ER oxidoreductin 1 (Ero1p), peroxiredoxin 4 (PRDX4), glutathione peroxidase 7/8 (GPx7/8), and ascorbate peroxidase (APX) predominantly degrade the ROS, ensuring ER stability (Rohli et al., 2022). However, excessive ROS can disrupt the redox balance in the ER, incapacitating disulfide bond isomerases and causing the accumulation of unfolded proteins, subsequently triggering ER stress (Kowalczyk et al., 2021).

Alcohol-induced disruption of lipid metabolism: implications for ER stress

Excessive alcohol consumption disrupts lipid metabolism by altering the expression and activity of pivotal enzymes involved in lipogenesis, fatty acid oxidation, and lipoprotein secretion (Bolatimi et al., 2023). For instance, alcohol can elevate the expression of the lipogenic gene-regulating transcription factor, sterol regulatory element-binding protein 1c (SREBP-1c), leading to increased lipogenesis and lipid accumulation in hepatocytes (Hyun et al., 2021). It is also impairs fatty acid oxidation by inhibiting the activity of peroxisome proliferator-activated receptor alpha (PPARα), a nuclear receptor involved in the regulation of genes controlling fatty acid β-oxidation (Zhang, Liu & Yang, 2023). Alcohol also affects lipid transport by modifying the secretion of lipoproteins, such as very-low-density lipoproteins (VLDL), which play a significant role in lipid export from hepatocytes (Wang et al., 2023). Studies utilizing C57BL/6J mice on long-term alcohol feeding and rapamycin-induced mTORC1 activation inhibition revealed that hepatic free fatty acids could initiate the mTORC1 signaling pathway, subsequently leading to ER stress (Guo et al., 2021). Moreover, accumulated lipids resulting from alcohol metabolism can compromise the integrity of ER membrane, thereby disturbing its regular functionality (Han & Kaufman, 2016). While the precise biological mechanism by which lipids instigate ER stress remains elusive, free fatty acids have been shown to obstruct protein folding, thereby precipitating ER stress (Lepretti et al., 2018). The lipid composition and quantity in the ER membrane could affect its fluidity and functionality, hindering the normal operation of membrane proteins and subsequently causing ER stress.

The ER serves as the primary site for lipid metabolism due to the presence of enzymes involved in lipid metabolism within the ER. Alcohol-induced dysregulation of lipid metabolism can lead to the accumulation of lipids within the ER membrane, causing ER stress and the activation of the UPR (Xia et al., 2020). It has been demonstrated that the UPR not only modulates ER homeostasis but also influences lipid metabolism. In the liver, lipogenesis relies on insulin-induced SREBP-1c activation. Recent findings revealed that CHI3L1 gene knock out led to decreased mRNA levels of the transcription factor SREBP1 in the livers of mice modeled with the Lieber-DeCarli diet, mitigating liver damage caused by the upregulation of SREBP-1 due to ALD. This highlights the potential therapeutic implications of SREBP-1 regulation for ALD (Lee et al., 2019). In addition, studies have shown that the overexpression of GRP78 in the mouse liver also repressed the cleavage of SREBP-1c and target gene expression of both SREBP-1c and SREBP-2, indicating that ER stress contributes to hepatic lipoatrophy in lab models (Kammoun et al., 2009). Furthermore, by specifically knocking out the DGAT1 gene in the liver of mice and subjecting them to chronic alcohol feeding, it was discerned that hepatocyte-specific deletion of DGAT1 led to lipid accumulation in the cells. This, in turn, activated ATF4, culminating in the induction of ER stress (Guo et al., 2021).

Er stress in the pathogenesis of ald

ALD encompasses conditions such as fatty liver, alcoholic hepatitis, and cirrhosis. Recent research suggests that ER stress and UPR may play significant roles in the pathogenesis of ALD. In ALD, persistent ER stress and UPR may lead to lipid metabolic disorder, hepatocyte inflammation, and even cell death, thereby exacerbating hepatic inflammation and fibrosis. This section provides a detailed overview of the mechanisms by which ER stress regulates hepatocyte lipid metabolism, inflammation, and apoptosis.

ER stress-induced hepatic lipid metabolic disorders

The enzymes required for lipid metabolism are widely distributed in the ER, making the liver a crucial hub for fat synthesis. Notably, the liver also dominates cholesterol synthesis in the body (Zhao et al., 2020). Under severe ER stress, the three main pathways of the UPR play crucial roles in the regulation of fatty degeneration. Each pathway may contribute to the progression of fatty liver degeneration by promoting fat breakdown, de novo fat generation, reducing fatty acid oxidation, and interfering with the secretion of lipoproteins and very low-density lipoproteins (Yu & Pajvani, 2023).

In liver fat metabolism, the IRE1α-XBP1 pathway significantly influences the assembly and secretion of VLDL and the de novo fats production (Ding et al., 2023). IRE1α has shown to inhibit vital metabolic transcription regulators, such as CCAAT/Enhancer Binding Proteins (C/EBP) β and δ, Peroxisome Proliferator Activated Receptor γ (PPARγ), and enzymes implicated in the biosynthesis of triglycerides. ER-stressed IRE1α-deficient mice exhibit severe fatty liver due to the hampered regulation of lipid synthesis and obstructed VLDL secretion (Zhang et al., 2011). Further, the Bax inhibitor-1 (Bl-1) gene has been found at lower levels in obese mice. Overexpression of Bl-1 seemed to stabilize lipid metabolism during the UPR by inhibiting the IRE1α endonuclease activity (Bailly-Maitre et al., 2010). Remarkably, XBP1 ablation in hepatocytes led to a significant reduction in cholesterol and triglycerides by curtailing liver lipid production (Lee et al., 2008). In hepatocytes, XBP1s interacts with the promoters of lipid metabolism-related genes (SCD, DGAT2), modulating their expression levels (Lee et al., 2008). Moreover, a negative feedback loop exists between XBP1 and IRE1α. The absence of the XBP1 triggers the activation of IRE1α, diminishing the degradation of downstream mRNAs related to lipid metabolism and inducing noticeable hypocholesterolemia in mice. Interestingly, either the deletion of XBP1 or the disruption of regulated IRE1-dependent decay (RIDD) can reverse the hypocholesterolemia observed in mice XBP1-deficient mice (So et al., 2012).

The PERK/eIF2α pathway also plays a vital role in liver fat metabolism. This pathway, when stimulated by palmitic acid ester-induced ER stress in hepatocytes, could promote steatosis by enhancing the expression of GADD153 or C/EBP Homologous Protein (CHOP), reducing the secretion of apolipoprotein B, and leading to the accumulation of triglycerides and cholesterol in hepatocytes (Guo et al., 2022). Under stress conditions, phosphorylated eIF2α can promote the expression of ATF4, which subsequently induces the expression of the transcription factor CHOP. This chain of events disrupts the function of C/EBP, inhibiting the expression of genes related to lipid metabolism, which then triggers disorders in fatty acid oxidation, lipoprotein secretion, and other lipid metabolic processes (Magee et al., 2022). In mice with ATF4 gene deletion, the expression of peroxisome proliferator-activated receptor-γ (PPAR-γ) in the liver is significantly reduced. PPAR-γ is a nuclear receptor that promotes fat production in the liver. The absence of ATF4 can weaken hepatic fat production by downregulating PPAR-γ, without affecting the production of hepatic triglycerides and fatty acid oxidation (Xiao et al., 2013). Knockout of the ATF4 gene leads to increased energy expenditure in mice and can inhibit diet-induced diabetes as well as hyperlipidemia and fatty liver (Seo et al., 2009). However, the basal level phosphorylation of eIF2α can prevent lipid accumulation caused by a direct challenge to ER stress because inactivation of eIF2α in mice can lead to tetracycline-induced fatty liver (Rutkowski et al., 2008).

ATF6, conversely, acts protectively against liver fatty lesions. Overexpression of a repressive version of ATF6 (dnATF6) elevates the susceptibility to hepatic steatosis in mice that manifest insulin resistance due to a high-fat diet (Chen et al., 2016). Additionally, a direct physical interaction is observed between ATF6 and PPARα. This interaction amplifies the transcriptional activity PPARα, subsequently activating target proteins of PPARα such as CPT1α and MCAD in hepatocytes (Flister et al., 2018). These proteins promote the oxidation of fatty acids in the liver, a process critical for controlling liver fat accumulation and energy balance. However, excessive fatty acid oxidation may lead to increased oxidative stress in the liver, potentially resulting in liver injury (Tu et al., 2020). Elevated ATF6 expression in the liver fosters hepatic fatty acids oxidation, providing protection for mice with insulin resistance caused by a high-fat diet from the impact of hepatic steatosis (Chen et al., 2016). Mice with ATF6α gene knockout exhibit significant liver dysfunction and fatty liver, and a notable accumulation of neutral (e.g., triglycerides and cholesterol) in liver. This accumulation stems from the reduction of related enzyme mRNA levels crucial for fatty acid β-oxidation, the instability of apolipoprotein B-100 that hampers the formation of very low-density lipoproteins, and lipid droplet synthesis triggered by the transcription of lipid differentiation-related proteins (Yamamoto et al., 2010). The absence of the ATF6α leads to blocked fatty acid oxidation, further promoting the early development of fatty liver (Tsitrina et al., 2023). When exposed to tunicamycin, ATF6α-deficient mice exhibit persistent express CHOP, inhibition of C/EBPα (CCAAT/enhancer-binding protein α), and develop hepatic steatosis (Lebeaupin et al., 2018). Both CHOP and C/EBPα are pivotal transcription factors that play a crucial regulatory role in liver lipid and glucose metabolism. Persistent CHOP expression might increase cellular sensitivity to stress, leading to cell death, while the inhibition of C/EBPα could imbalance liver lipid metabolism and glucose metabolism, eventually leading to hepatic steatosis (Lebeaupin et al., 2018). In addition, ATF6 can inhibit the transcriptional activation of SREBP2, thereby suppressing the lipid-generating effects of SREBP2 in hepatic cells (Zeng et al., 2004).

The UPR acts as a counteractive mechanism to ER stress, striving to maintain the homeostasis of the ER. Nonetheless, sustained ER stress can activate UPR for a prolonged period, leading to lipid metabolism disturbances. Each UPR pathway plays a distinct role in the lipid metabolic disorder induced by ER stress, and the interactions and specific mechanisms among these pathways still require further in-depth research. Overall, UPR is indispensable for hepatic lipid metabolism. Deepening our understanding of its operations and regulatory mechanisms not only demystifies the intricacies of lipid metabolism but also paves the way for innovative treatments for hepatic diseases related to lipid metabolism.

ER stress and the hepatic inflammation

ER stress is associated with several cellular inflammatory response pathways. These include the activation of NF-kB, JNK, ROS, Interleukin-6 (IL-6), and tumor necrosis factor-α (TNF-α) (Malhi & Kaufman, 2011). Notably, NF-κB activation plays a predominant role in inflammatory responses, serving as a crucial mediator in hepatocellular damage, fibrosis, and the progression of hepatocellular carcinoma. Research indicates that the phosphorylation of eIF2α is vitally important in driving NF-kB transcriptional activity. During ER stress, the PERK pathway can activate NF-kB by inhibiting the release of protein IkappaB (Du et al., 2022). Pathogen-generated lipopolysaccharides in the intestine may exert toxic effects on hepatocytes, and activate the NFkB-mediated inflammatory response in stem cells. This response bears a dual role, promoting inflammation while opposing apoptosis. This highlights a pivotal regulatory function of NF-kB-mediated inflammatory response in hepatocytes. In this context, a mild up-regulation of NF-kB can counteract hepatitis by inhibiting hepatocellular apoptosis via inflammatory responses. However, excessive up-regulation of NF-kB could facilitate the release of inflammatory factors, intensifying the severity of hepatitis (Luedde & Schwabe, 2011). In addition, NF-kB can also be activated through the phosphorylation of Akt in the ATF6 pathway (Yamazaki et al., 2009), as well as through the CHOP pathway (Willy et al., 2015).

ER stress and the hepatocyte apoptosis

ALD, a consequence of chronic and excessive alcohol consumption, encompasses a spectrum of liver conditions, encompasses a spectrum of liver conditions, from fatty liver and alcoholic hepatitis to cirrhosis, and even liver cancer. Cell apoptosis is a significant factor in the progression of ALD, as depicted in Fig. 3. The primary pathways of cell apoptosis in ALD involve the extrinsic pathway, the mitochondrial pathway, and the ER stress pathway. Notably, the ER stress-induced apoptosis pathway further bifurcates into three significant routes (Beilankouhi et al., 2023): the IRE1α signaling pathway, CHOP-induced apoptosis, and activation of the caspase-12 pathway.

Figure 3 Schematic representation of the pathways involved in ALD progression and the pivotal role of ER stress in cell apoptosis.

The role of IRE1α pathway in regulating the hepatocyte apoptosis

Under high ER stress, there is evidence indicating that IRE1 may promote the non-specific degradation of membrane-associated mRNA through mechanism termed regulated IRE1-dependent decay (RIDD) (Hollien & Weissman, 2006). While previous reports have identified RIDD’s involvement in the degradation of ER stress-related mRNA, limiting the synthesis of nascent proteins (Zhang et al., 2011) to relieve ER stress, it also seems to be participated in the apoptotic pathway under acute ER stress conditions. RIDD cleaves RNA at a consensus site similar to XBP1, but its activity is distinct from XBP1 mRNA splicing. It holds the capacity to either preserve ER equilibrium or prompt cellular apoptosis (Maurel et al., 2014). Additionally, studies have found that mice with BAX and BAK gene knockout showed abnormal response to tunicamycin-induced ER stress, accompanied by extensive tissue damage. The expression of IRE1 substrate X-box binding protein 1 and its target genes were also reduced. Co-immunoprecipitation experiments revealed the interaction between IRE1α and BAK and BAX proteins, essential players in the mitochondrial apoptotic pathway, underscoring the potential involvement of BAX and BAK in activating the IRE1 signaling pathway and liver apoptosis (Hetz et al., 2006).

Within the IRE1 pathway, phosphorylated IRE1 also binds to the receptor protein tumor necrosis factor receptor-associated factor 2 (TRAF2). This interaction subsequently promotes a series of phosphorylation events, leading to the activation of Jun amino-terminal kinase (JNK) (Urano et al., 2000). Prolonged JNK activation can drive liver cell demise (Basha et al., 2023). Once activated, phosphorylated JNK activates the pro-apoptotic protein Bim while deactivating the anti-apoptotic protein Bcl2, orchestrating cell apoptosis. Additionally, overexpressed JNK can compromise the ER membrane integrity, causing efflux of Ca2+ ions. This sequence activates Caspase12 via proteases, further pushing the cell towards apoptosis (Stillger et al., 2023). Wu et al. (2020) conducted a study where mice were orally administered with 10 mg/kg of copper sulfate, resulting in significant ER stress, elevated gene expression levels in the JNK and Caspase12 signaling pathways, and the onset of liver cell apoptosis. This study further elucidates the relationship between the JNK and Caspase12 signaling pathways, ER stress, and liver cell apoptosis.

The role of CHOP pathway in regulating the hepatocyte apoptosis

CHOP, an integral part of the C/EBPs protein family, functions as a transcriptional regulator that facilitates apoptosis under the influence of ER stress. Investigations utilizing deletion mutations have elucidated the pivotal function of the C-terminal bZIP structural domain in CHOP-mediated apoptosis initiation (Ubeda et al., 1996). The apoptotic cascade mediated by CHOP exhibits significant association with the processes involving PERK, ATF6, and IRE1 (Oyadomari & Mori, 2004). During instances of ER stress, phosphorylated eIF2α enhances the transcription of ATF4, leading to the upregulation of genes such as CHOP, GADD34 (Michel et al., 2015). Notably, mice lacking PERK and ATF4 demonstrate an inability to activate -mediated apoptosis during ER stress (Harding et al., 2003). The eIF2α pathway triggers the expression of PERK and ATF4, amplifying protein synthesis, which in turn leads to ATP exhaustion, ROS generation, and subsequent apoptosis (Han et al., 2013). Within the ATF6 pathway, the activated migrates to the cell nucleus, promoting the transcription of several UPR-associated genes, including CHOP and XBP1 (Zimmermann et al., 2023). In the IRE1 pathway, XBP1, serving as a downstream transcription factor, can further augment the expression of the CHOP gene (Liu, Zhao & Rutkowski, 2023). Additionally, the phosphorylation cascade triggered by IRE1 encompasses not only JNK but also p38 mitogen-activated protein kinase (p38 MAPK). The latter is implicated in further phosphorylation of serine located at positions 78 and 81 of CHOP, thereby inducing apoptosis (Ron & Hubbard, 2008).

The pathways through which CHOP induces cell apoptosis include: (1) Inhibition of the expression of anti-apoptotic proteins in the BCL-2 family. McCullough et al. (2001) found that cell lines with heightened CHOP expression were especially vulnerable to ER stress, exhibiting a decrease in Bcl-2 expression. Restoring Bcl-2 expression in these cells counteracted the CHOP-induced increase in ROS and cell apoptosis. How CHOP regulates the promoter of Bcl-2 remains unclear, but studies have reported that CHOP can facilitate its transport within the nucleus through interaction with the bZIP protein C/EBP beta subtype (LIP) in the cytoplasm and nucleus. Notably, mouse embryonic fibroblasts expressing LIP have shown enhanced ER-induced apoptosis (Chiribau et al., 2010). CHOP can cause cell apoptosis by reducing the expression of Bcl-2, depleting cellular thiols, leading to ROS production, and further disrupting oxidative balance (McCullough et al., 2001). (2) CHOP induces cell apoptosis through ROS. CHOP can oxidize the ER lumen via ER oxidase1α (ERO1α) (Fujii, Ushioda & Nagata, 2023). In the absence of ER stress, the oxidation of ERO1α promotes the formation of disulfide bonds in newly entered proteins in the ER. However, persistent ER stress can induce hyper-oxidation of the ER environment, leading to cell death. It was found that cells lacking CHOP can reduce oxidative damage and achieve higher cell survival rates by reducing the expression levels of the ERO1α gene and ROS-related stress genes (Deng et al., 2023). Furthermore, CHOP gene knockout mice exhibited a significant reduction in cell apoptosis when exposed to agents that impair ER function (Zinszner et al., 1998). (3) CHOP-GADD34-elF2α pathway induced cell apoptosis. In the UPR, cells primarily mitigate ER stress and cell apoptosis by reducing protein translation rates through the phosphorylation of elF2α (Hanson et al., 2022). Under these conditions, CHOP promotes cell apoptosis by enhancing GADD34 transcription, subsequently facilitating the dephosphorylation of the 51st serine of phosphorylated elF2α, thereby reinvigorating protein transcription. Studies of gene knockout in mice by Marciniak et al. (2004) found that both GADD34 gene and CHOP knockout mice were resistant to tunicamycin-induced ER stress, reinforcing the aforementioned hypothesis.

The role of caspase-12 in regulating the hepatocyte apoptosis

Caspases, a group of cysteine proteases, are integral components in the orchestration of apoptosis. Within this family, the role of caspase-12 has gained significant attention due to its participation in the apoptosis of hepatocytes, particularly in instances of cell demise triggered by ER stress. Under normal circumstances, Tumor Necrosis Factor Receptor-Associated Factor 2 (TRAF2) forms a complex with the caspase-12 precursor. However, under sustained ER stress, the caspase-12 precursor dissociates from TRAF2, thereby activating caspase-12 (Nakagawa et al., 2000). Under sustained ER stress, caspase-7 cleaves the caspase-12 precursor at the Asp94 and Asp31 sites, thus activating caspase-12. Activated caspase-12 then activates caspase-9, which in turn activates caspase-3, ultimately leading to the induction of cell apoptosis (Pal et al., 2015). Additionally, caspase-12 can also be cleaved by the calcium-regulated enzyme calpain, activating procaspase-12 to generate active caspase-12 (Bonsignore, Martinotti & Ranzato, 2023). In addition, research by Ding et al. (2022) demonstrated that Guanylate Binding Protein 5 (GBP5), a member of the guanosine triphosphate binding protein family, exhibits abnormally elevated expression levels in cases of liver damage and validation. Furthermore, the inhibition of calpain activity or caspase-3 can prevent GBP5-induced cell apoptosis (Ding et al., 2022). This implies a key role of GBP5 in regulating the caspase-12 cell apoptosis pathway.

ER stress and autophagy in ALD

It is reported that in the zebrafish model of ALD, the alcohol metabolism leads to impaired ER function and activation of downstream targets of the UPR (Tsedensodnom et al., 2013). When hepatocytes face prolonged ER stress, the UPR activation alone is inadequate to mitigate the stress, prompting the onset of autophagy to preserve ER stability (Senft & Ronai, 2015). In the UPR-triggered autophagy, PERK plays a pivotal role by activating autophagy-related genes through the phosphorylation of eIFα. Kouroku et al. (2007) found that mutations in the phosphorylation sites of eIFα or knockout of the PERK gene diminish the cellular autophagy induced by ER stress. Additionally, in the IRE1 pathway, the binding of TRAF2 to IRE1 activates JNK, which further phosphorylates Bcl-2. This leads to the release of the autophagy-regulating protein Beclin-1 from Bcl-2, activating the phosphoinositide-3-kinase (PI3K) complex and autophagy (Deegan et al., 2013). Moreover, C/EBP-β is also implicated in this process (Parzych & Klionsky, 2014). In a study led by Lin et al. (2013) an enhancement in autophagic flux was observed in mice subjected to extended ethanol feeding. The investigation involved the administration of carbamazepine (an autophagy activator) to mice under both acute and chronic ethanol conditions. Notably, carbamazepine was found to alleviate hepatitis and hepatic injury in ALD mice by augmenting autophagic activity (Lin et al., 2013). Specific knockout of the DGAT1 gene in mouse hepatocytes revealed that these DGAT1-deficient mice exhibited an upregulation of ER stress and a downturn in autophagy mediated by the LAMP2 pathway, which consequently led to liver damage (Guo et al., 2021). In a related study, Chao et al. (2018) reported that mice with a knockout of the Transcription Factor EB gene, a pivotal regulator for the transcription of autophagy-required genes, manifested exacerbated steatosis post-ethanol treatment compared to their control counterparts. It is evident from these findings that autophagy plays a protective role in the progression of ALD, wherein alcohol intake can stimulate cellular autophagy as a self-protective response (Ding et al., 2010). However, chronic alcohol consumption appears to suppress autophagy (Xia et al., 2020), thereby exacerbating hepatic injury.

Compounds regulating ER stress in ALD

In summary, ER stress is central to the pathogenesis of. Our comprehension of the three pathways initiated by ER stress, coupled with insights into how alcohol induces such stress, provides fresh perspectives on the pathological mechanisms underlying ALD. This knowledge paves the way for the development of innovative treatment strategies, especially drugs that specifically target ER stress. In reference to the aforementioned IRE1α, PERK, and ATF6 pathways, the final section of this review encapsulates some known and potential drugs, as detailed in Table 1. These drugs primarily influence ALD-related conditions, including fatty liver, alcoholic hepatitis, and liver cancer, through the modulation of the aforementioned pathways, either mitigating or intensifying ER stress. By deepening our understanding of the mechanisms of action of these drugs, we aspire to identify more targeted and effective treatment methods to better combat the pathogenesis of ALD.

Table 1 Compounds that target to the ER stress related pathways in ALD.

Targets	Name	Effects	Liver disease	Reference	
IRE1α	STF-083010	Inhibition the endonuclease activity of IRE1	Liver fibrosis	Liu, Zhao & Rutkowski (2023)	
	4μ8c	Inhibition the RNase activity of IRE1α	Steatohepatitis	Ron & Hubbard (2008)	
	MKC-3946	Blocking of XBP1 splicing	–	McCullough et al. (2001)	
	KIRA6	Inhibit IRE1α’s RNase by breaking oligomers	–	Chiribau et al. (2010)	
	Selonserib	Reduces inflammation	Liver fibrosis	Fujii, Ushioda & Nagata (2023)	
PERK	GSK2656157	Inhibits PERK activation	Liver fibrosis	Deng et al. (2023)	
	GSK2656157	Reduces PERK autophosphorylation	HCC	Zinszner et al. (1998)	
	FGF21	Inhibits PERK-ATF4 pathway	Steatohepatitis	Hanson et al. (2022)	
	ISRIB	Inhibits eIF2α phosphorylation	–	Marciniak et al. (2004)	
ATF6	Ceapins	Block ATF6α signaling in response to ER stress	–	Nakagawa et al. (2000)	
Others	Adenoviral vector-GRP78	Overexpression of GRP78	Steatohepatitis	Pal et al. (2015)	
	4-PBA	Promotes protein folding and activatesautophagy	ALD	Bonsignore, Martinotti & Ranzato (2023)	
	Mg132	Proteasome inhibitor	ALD, HCC	Ding et al. (2022)	
	Bortezomib	Inhibition of UPR and amplification of ER stress	HCC	Tsedensodnom et al. (2013)	
	VCPI/OV combination therapies	VCP inhibitors cooperated with M1 virus-suppressed XBP1 pathway and triggered irresolvable ER stress	HCC	Senft & Ronai (2015)	
	Melatonin	Induction of CHOP apoptosis pathway	HCC	Kouroku et al. (2007)	
	Resveratrol	Enhance of XBP1s Splicing and CHOP expression	HCC	Deegan et al. (2013)	
	TUDCA	Function as a chemical chaperone and reduce ER stress	ALD	Parzych & Klionsky (2014)	

Conclusions

Intake of ethanol leads to an elevated production of ROS in hepatocytes, intensifying the onset of ER stress. Mouse models indicate that this escalation is attributed to the decline in SOD levels and the glutathione/oxidized GSH ratio due to the metabolism of ethanol to acetaldehyde, combined with an increased activity of ethanol-metabolizing enzymes (Li et al., 2019; Farfán Labonne et al., 2009; Li et al., 2017). Such findings underscore the potential of ROS reduction in the prevention and treatment of ALD. Additionally, ethanol can enhance lipid production by modulating the activity of SREBP-1c (Guo et al., 2021). The accumulation of lipids in hepatocytes not only suppresses autophagy but also exacerbates ER stress (Lee et al., 2019), further contributing to liver damage. The activation of autophagy has been demonstrated to be a promising therapeutic strategy for ALD (Nissar et al., 2017). Prolonged ER stress that exceeds the regulatory scope of the UPR can trigger disorders in lipid metabolism, inflammation, and apoptosis in hepatic cells. The pathological process of ALD is highly complex encompassing more than just ER stress—it also involves the modulation of several signaling pathways, including miRNA. Understanding the interconnections between these varied mechanisms and their ties with ER stress will be pivotal in future research endeavors.

ADDITIONAL INFORMATION AND DECLARATIONS.

Competing Interests

Author Contributions

Data Availability

The authors declare that they have no competing interests.

Man Na conceived and designed the experiments, analyzed the data, prepared figures and/or tables, authored or reviewed drafts of the article, and approved the final draft.

Xingbiao Yang performed the experiments, analyzed the data, authored or reviewed drafts of the article, and approved the final draft.

Yongkun Deng performed the experiments, analyzed the data, authored or reviewed drafts of the article, and approved the final draft.

Zhaoheng Yin analyzed the data, authored or reviewed drafts of the article, and approved the final draft.

Mingwei Li conceived and designed the experiments, performed the experiments, analyzed the data, authored or reviewed drafts of the article, and approved the final draft.

The following information was supplied regarding data availability:

This is a literature review.

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
