# Peer review of "Endoplasmic reticulum stress in the pathogenesis of alcoholic liver disease"

_PeerJ, doi:10.7717/peerj.16398_

## Round 0.1 · original submission · Major Revisions

1. As suggested by the reviewers, the authors should include relevant references in support of the statement they made in different sections.

2. Since this article focused on the role of ER stress on the pathogenesis in alcoholic liver diseases, it would be important to describe alcohol metabolism briefly.

3. Please make the editorial corrections, typing errors, and missing references throughout the manuscript to improve its quality.

·

Basic reporting

NO comment

Experimental design

No comment

Validity of the findings

No comment

Additional comments

The review article titled “Endoplasmic reticulum stress in the pathogenesis of alcoholic liver disease” is well written and engaging. There are a couple of review articles published related to ER and liver diseases. To make the current article more interesting I have the following comments:


1. It would be very helpful if the authors show schematic between the relationship of ER stress and the hepatocyte apoptosis with respect to ALD.
2. It will broaden the review perspective if the authors include mouse model related study. And also can dedicate an section of discussion on this aspect.
3. A section devoted to ER stress and autophagy would be very interesting.
4. The figure 1 is very well depicted. It would be helpful if the authors provide more details on the mechanism of ethanol metabolism in the liver.

Reviewer 2 ·

Basic reporting

no comment

Experimental design

no comment

Validity of the findings

no comment

Additional comments

The paper introduces ER stress and the unfolded protein response (UPR), highlighting their interrelation. It also explores how alcohol triggers ER stress by generating reactive oxygen species, impacting lipid synthesis, and influencing ER protein production. These factors collectively worsen ER stress, potentially leading to lipid metabolism disorders, inflammation, and hepatic cell apoptosis.
I recommend improving the language throughout the manuscript and adding references wherever it is needed and missing. I have specified some suggestions.

1. In figure 1, correct liqid to lipid
2. In lin 48, functions of hepatocytes? Explain
3. Line 66, add references
4. Line 76, enhances ERAD? Full form/describe?
5. Line 78, in vitro, use italics fonts
6. Line 95, mice research how it is relevant? Improve writing here, what is Tunicamycin (Tunicamycin (TM) is an inhibitor of glycosylation??)
7. Line 98, The various pathways of UPR are illustrated in Fig2, correct this sentence and the next one. “The” is not required.
8. Line 105, “Alcohol consumption, specifically ethanol, can lead to ER stress through several mechanisms.” Describe here in short or say describes below.
9. In line 107, various health issues? Such as?
10. Line 115, Microsomal Ethanol-Oxidizing System (MEOS)? Why the first letter in each word is capitalized?
11. Line 116-121 references?
12. Line 125 reference?
13. Line 132, Ero1, ER oxireductin?
14. Line 155, “lipid metabolism within the ER in humans.” only in humans?
15. Line 185, b1-1? I think should be Bl-1 (Bax inhibitor gene)
16. Line 195, RIDD? Explain? Is it regulated IRE-1 dependent decay?
17. Line 214, reference?
18. Line 217, reference?
19. Wherever each word is used first in the paper make sure you either describe it necessary or write full form throughout the manuscript (example, CHOP in line 232 and it has been used earlier in the text)
20. Line 265-270, references?

---

## Round 0.2 · accepted · Accept

Thank you for addressing all the comments which improved the quality of the manuscript.

·

Basic reporting

The context of the paper is clear with good references.

Experimental design

NA

Validity of the findings

NA

Additional comments

The authors has answered all my concerns satisfactorily.

Reviewer 2 ·

Basic reporting

no comment

Experimental design

no comment

Validity of the findings

no comment

Additional comments

I am writing to express my positive recommendation for the publication of the manuscript titled Endoplasmic reticulum stress in the pathogenesis of alcoholic liver disease submitted to PeerJ Life and Environment. After carefully reviewing the revised version, I am pleased to report that the authors have diligently incorporated all the suggested revisions, significantly improving the quality of the manuscript.
Considering the substantial efforts made by the authors to address the comments and enhance the overall content, I believe that the paper is now ready for publication. Therefore, I recommend accepting it for publication in PeerJ Life and Environment.